# Generic Methods for Simultaneous Analysis of Four Direct Oral Anticoagulants in Human Plasma and Urine by Ultra-High Performance Liquid Chromatography-Tandem Mass Spectrometry

**DOI:** 10.3390/molecules28052254

**Published:** 2023-02-28

**Authors:** Jian-Wei Ren, Xin Zheng, Xiao-Hong Han

**Affiliations:** Clinical Pharmacology Research Center, Peking Union Medical College Hospital, Beijing Key Laboratory of Clinical PK & PD Investigation for Innovative Drugs, NMPA Key Laboratory for Clinical Research and Evaluation of Drug, State Key Laboratory of Complex Severe and Rare Diseases, Chinese Academy of Medical Sciences & Peking Union Medical College, Beijing 100730, China

**Keywords:** DOACs, human plasma, urine, UPLC-MS/MS

## Abstract

The new direct oral anticoagulants (DOACs) are increasingly used to treat and prevent thromboembolic disorders, and monitoring concentrations may be valuable in some special scenarios to prevent clinical adverse events. This study aimed to develop generic methods for the rapid and simultaneous analysis of four DOACs in human plasma and urine. Protein precipitation and one-step dilution were used to prepare the plasma and urine; the extracts were injected to ultra-high performance liquid chromatography-tandem mass spectrometry (UPLC-MS/MS) for analysis. Chromatographic separation was performed on an Acquity™ UPLC BEH C_18_ column (2.1 × 50 mm, 1.7 μm) with gradient elution of 7 min. A triple quadrupole tandem mass spectrometer with an electrospray ionization source was employed to analyze DOACs in a positive ion mode. The methods showed great linearity in the plasma (1~500 ng/mL) and urine (10~10,000 ng/mL) for all analytes (R^2^ ≥ 0.99). The intra- and inter-day precision and accuracy were within acceptance criteria. The matrix effect and extraction recovery were 86.5~97.5% and 93.5~104.7% in the plasma, while 97.0~101.9% and 85.1~99.5% in the urine. The stability of samples during the routine preparation and storage were within the acceptance criteria of less than ±15%. The methods developed were accurate, reliable, and simple for the rapid and simultaneous measurement of four DOACs in human plasma and urine, and successfully applied to patients and subjects with DOACs therapy for anticoagulant activity assessment.

## 1. Introduction

In recent years, thrombotic diseases have become one of the major reasons that threaten human health around the world [1]. Since 2010, the new direct oral anticoagulants (DOACs), dabigatran, rivaroxaban, apixaban, edoxaban, and betrixaban, have been developed and approved successively for first-line therapy or the prevention of stroke, thrombosis, and pulmonary embolism. Due to the predictable pharmacological response, the DOACs are usually administered in a fixed-dose without routine laboratory monitoring [2]. However, some special factors, such as extreme age, severe renal impairment, and co-administration, may lead to dangerous DOACs plasma levels [2,3,4]. Therefore, drug monitoring and analysis may be meaningful in some special scenarios.

At present, several methodologies based on liquid chromatography mass spectrometry (LC-MS) have been established for the quantification of DOACs [5]. To our knowledge, the majority of them were single-analyte approaches and no longer met the current requirements of laboratory monitoring. The multi-analyte assay avoids the frequent switching of the single-analyte approach, and the uniform sample preparation eliminates cumbersome procedures, which is an inevitable trend for the clinical laboratory monitoring in the future. The multiple-analyte assays previously reported involved labor-intensive sample preparations, and baseline separation was never observed, which caused mutual interference [6,7,8]. Urine samples are conveniently collected without invasive injury and the DOACs levels in the urine correlate well with the concentrations in the plasma, suggesting that urine may be the surrogate matrix for plasma [9].

In the present study, generic multiple-analyte approaches were developed for the rapid and simultaneous quantification of apixaban, edoxaban, betrixaban, and rivaroxaban in human plasma and urine, and were successfully applied to patients and subjects with DOACs therapy for drug monitoring and anticoagulant activity assessment.

## 2. Results

### 2.1. Method Development

#### 2.1.1. Chromatography and Mass Spectrometry

Different columns (Acquity™ UPLC BEH C_18_, Acquity™ UPLC CSH C_18_, Acquity™ UPLC BEH Phenyl, 2.1 × 50 mm 1.7 μm, Waters, Eschborn, Germany) were elementarily investigated, and the best separation and most intense [M + H]^+^ ions were achieved on an Acquity™ UPLC BEH C_18_ column with the gradient program. Baseline separation was observed from Figure 1, which illustrated the typical extracted ion chromatograms of all analytes. The optimized ionization parameters were exhibited in Table 1.

#### 2.1.2. Sample Preparation

Labor-intensive and time-consuming operations are not undesirable in clinical laboratories. In this study, the plasma samples were precipitated with acetonitrile, then diluted with mobile phase A. For urine samples, the one-step dilution had adequate sensitivity, which just needed a few minutes. It had been tested that the great mass spectral response and mild ion suppression of all analytes were observed under the sample preparation.

### 2.2. Method Validation

#### 2.2.1. Linearity

Excellent linearity of all analytes was observed in the range of 1~500 ng/mL in the plasma and 10~10,000 ng/mL in the urine, and the correlation coefficients (R^2^) were between 0.9956 and 0.9993. At least 75% of the calibration standard conformed to the criterion in 2.6.1. The detailed result was listed in Table 2.

#### 2.2.2. Selectivity

As demonstrated in Figure 2 and Figure 3, there was no evident interference from the blank plasma and urine observed at the retention times for all analytes and their internal standards.

#### 2.2.3. Precision and Accuracy

The intra- and inter-day precision and accuracy results were depicted in Table 3. Both the relative standard deviation (RSD) and relative error (RE) at the lower limit of quantitation (LLOQ), low quality control (LQC), medium quality control (MQC), and high quality control (HQC) were within the acceptance criteria, which indicated that the developed method was reliable and reproducible.

#### 2.2.4. Matrix Effect and Extraction Recovery

The matrix effect and extraction recovery data of all analytes are listed in Table 4. The result met the acceptance criterion. The result revealed that the ion suppression from human plasma and urine was negligible.

#### 2.2.5. Effect of Hemolysis and Hyperlipidemia in Plasma

The precision (RSD) and accuracy (RE) were within the acceptance criteria (shown in Table 5), which entailed that slight hemolysis and hyperlipidemia had no influence on analytes quantification.

#### 2.2.6. Dilution Integrity

The precision (RSD) of the dilution quality controls (DQCs) ranged from 1.3% to 2.5% and the accuracy (RE) ranged from 3.3% to 4.8% for all analytes with a dilution factor of 10. For the plasma and urine samples above the upper limit of quantification, it was stable and accurate to analyze after the 10-fold dilution with the pooled plasma or urine.

#### 2.2.7. Stability

The results of the short-term stability, reinject stability, autosampler stability, freeze-thaw stability, and long-term stability were shown in Table 6 and Table 7, and the accuracy (RE) and precision (RSD) were within the acceptance criteria. The results revealed good stability for all analytes during the routine preparation and storage processes.

#### 2.2.8. Carry-Over

No peaks of analytes and internal standards were observed at their respective retention times in the blank plasma and urine following the upper limit of quantitation, which demonstrated the negligible carryover.

### 2.3. Patient Samples Analysis

The developed and validated UPLC–MS/MS method was successfully applied to human plasma and urine. The UPLC-MS/MS chromatograms of DOACs in the plasma and urine sample were presented in Figure 4, which confirmed that our method qualified to analyze the plasmas and urine samples of the patients and healthy subjects. For the two patients treated with rivaroxaban, we plotted the abbreviated concentration-time curve (Figure 5). The peak concentrations of rivaroxaban in the plasma (226~284 ng/mL) were observed within 2~4 h of administration and the 24 h urinary excretion rate was approximately 15.0%~19.6%, which was consistent with those reported in previous research [10,11,12]. For the subjects receiving apixaban and edoxaban, the brief concentration-time curves were illustrated in Figure 3. Similar to rivaroxaban, apixaban and edoxaban reached their peak concentrations (225 and 70.2 ng/mL) in the plasma 2~4 h after oral administration with the trough concentrations of 4.24 and 4.27 ng/mL, respectively. Additionally, apixaban was excreted approximately 9.6% in the urine within 4 h after administration, and approximately 14.4% for edoxaban. In 2018, the International Council for Standardization in Haematology (ICSH) issued the expected peak and trough DOAC concentrations in patients, which could be a potentially therapeutic range. The peak and trough concentrations detected in this study were within the range, which indicated the appropriate anticoagulant activity in patients and subjects and low risk of adverse clinical outcomes.

## 3. Discussion

The DOACs generally do not require routine laboratory monitoring; whereas, in some special situations, monitoring may be helpful to ensure the required exposure. The drug-drug interaction was an important cause of adverse outcomes in the real world. Recently, several studies have reported the increased major bleeding risk when co-administered with 5-phosphodiesterase inhibitors, antiepileptics, and antivirals [13,14,15,16]. In addition, the clearance of DOACs in the human body was associated with the P-glycoprotein and the breast cancer resistance protein, which have single nucleotide polymorphism, and the patients with heterozygous mutation and homozygous mutation have more rapid clearance than homozygous wild type [17]. Recently, Zhang Dan et al. investigated the relationship between the glomerular filtration rate and total bilirubin, and the plasma DOACs concentration in the elderly with non-valvular atrial fibrillation, and a low dose was recommended for elderly patients to prevent bleeding events [17]. Therefore, drug monitoring may be beneficial in patients with impaired renal and hepatic function, and in patients requiring co-administration.

In 2012, the British Committee for Standards in Haematology (BCSH) published guidance for dabigatran and rivaroxaban therapy; patients taking rivaroxaban within 24 h before surgery, patients with renal failure, patients with bleeding, patients with overdose, and patients with thrombosis were recommended to monitor drug concentrations during treatment [18]. Furthermore, in 2014, the recommended patients increased to 13 groups, demonstrating the importance of drug monitoring [19]. Interestingly, similar recommendations have been made by the King Thrombosis Centre in Denmark, which places more emphasis on the extreme weight patients (body weight more than 120 kg or less than 40 kg) and patients with severe renal impairment (renal creatinine clearance < 30 mL/min), compared to BCSH [20]. In 2018, the ICSH issued the recommendations for the laboratory monitoring of DOACs and listed the anticipated peak and trough concentrations of dabigatran, rivaroxaban, apixaban, and edoxaban for the prevention and treatment of stroke, pulmonary embolism, non-valvular atrial fibrillation, and venous thromboembolism, which was instructive for the interpretation of monitoring results [21]. In the real world, approximately 1%~3% of patients with long-term DOACs therapy each year experience bleeding and thromboembolism events, and inappropriate plasma DOACs concentrations may be the cause [22,23,24]. The association between the plasma concentrations of edoxaban and adverse outcomes was studied, and the stroke and thromboembolism were prone to occur in patients with low trough concentrations, while bleeding events occurred more frequently in the patients with high trough concentrations [2].

In the multi-analyte approaches developed by Kathrin et al. [6] and Vítězslav et al. [25], the chromatographic peaks of apixaban, rivaroxaban, and edoxaban were completely overlapping, which may account for the unreliable quantification. In this study, the baseline separation of all analytes was evidently observed in Figure 1. The method established by Kathrin et al. [6] was the only multi-analyte approach including betrexaban, and the carryover of betrixaban exceeded the acceptance criteria, which made it necessary to inject blank samples between the adjacent samples to avoid interference, while carryover was never found in our method. Importantly, the influence of hemolysis and hyperlipidemia in the plasma on the multi-analyte quantification was hardly investigated previously, which might bring about unreliable results, and the results in Table 5 indicated that hemolysis and hyperlipidemia had no effect on the multi-analyte simultaneous quantification.

It was reported that approximately 30~80% of DOACs in the systemic circulation were excreted in the urine [5]. Urine samples are collected conveniently without invasive injury and additional bleeding risk during the anticoagulant therapy. The monitoring of the urine samples may simplify the decision-making process in patients with acute stroke, severe major bleeding, and emergency surgery, who need rapid and accurate information on administered DOACs. A recent study confirmed the great correlation of DOACs levels in the plasma and urine, and urine was reliable to estimate relevant DOACs exposure in patients and had the potential to accelerate the decision-making in emergencies [9]. For children, the elderly, and patients with difficulty in blood collection, urine may be a reasonable surrogate matrix. At present, only Tzu-Yu Panet al. [26] have reported the multi-analyte approach in urine. They developed an ultrasound-assisted salt-induced liquid-liquid microextraction to extract DOACs from urine, which was a four-step procedure and not as simple and rapid as the one-step dilution we developed. Moreover, multiple factors, including extraction solvent, volume of extractant, added salts, and the pH of the extraction solution, affected the extraction recovery, which decreased the method reliability. Recently, several qualitative in vitro diagnostic DOACs dipsticks have been developed for the point-of-care (POC) test in emergency situations with accuracy, sensitivities, and specificities more than 95%, and the results can be obtained visually within 10 min [27,28,29]. However, the color of urine, the amount of urine, and reducing substances in the urine may lead to false negative results. In our method, the urine samples are directly injected for analysis after a one-step dilution, and qualitative and quantitative analysis can be completed simultaneously within 10 min after receiving the urine.

Furthermore, the method developed in this study may be valuable for patients who are comatose, unconscious, or unable to inform clinicians of their anticoagulation therapy, as well as for forensic identification. The detection of four factor Xa inhibitors can be performed in a single run rapidly, allowing doctors to administer the corresponding antidote in time. In a word, the method established in this study showed great superiority in reliability, analytical efficiency, and operability compared with those previously reported.

According to the Recommendations for Laboratory Measurement of Direct Oral Anticoagulants published by ICSH, DOACs metabolites with anticoagulant activity need to be quantified at the same time [21]. After oral intake, edoxaban is transformed to several metabolites (M1, M2, M3, M4, M5, M6, and M8). Among them, M4, M6, and M8 have anticoagulant activity, which merely accounts for less than 10% of the total anticoagulant activity [30]. Hence, the active metabolite of edoxaban was not involved in this study, which might lead to an underestimation of the anticoagulant activity of edoxaban. In addition, plasma and urine samples from patients taking betrixaban have not be collected and analyzed, since betrixaban is not yet available in mainland China. Due to the limited plasma and urine samples, we did not have enough data to verify the correlation between the two, which required further research.

## 4. Materials and Methods

### 4.1. Chemicals and Reagents

Apixaban, betrixaban, rivaroxaban, edoxaban, and ^2^H_4_-rivaroxaban were purchased from MedChemExpress (Monmouth Junction, NJ, USA). ^2^H_3_-Apixaban, ^2^H_6_-edoxaban, and ^2^H_6_-betrixaban were obtained from SHANGHAI ZZBIO CO., LTD (Shanghai, China). HPLC-grade methanol and acetonitrile were purchased from Fisher, USA. Ammonium acetate (A.R. grade), ammonium formate (A.R. grade), and ammonium hydroxide (A.R. grade) were bought from Sinopharm Chemical Reagent Co., Ltd. (Beijing, China). Sigma-Aldrich Corp. (St. Louis, MO, USA) supplied the formic acid (A.R. grade) and dimethyl sulfoxide (HPLC-grade). Deionized water was purified with a Milli-Q^®^ Ultrapure water system (Millipore Corporation, Bedford, MA, USA).

### 4.2. Instruments

An AB SCIEX API 4000 triple quadrupole tandem mass spectrometer (AB SCIEX, Toronto, Canada) and a Shimadzu HPLC equipped with two solvent delivery units (LC-20AD XR), communication bus module (CBM-20A), autosampler (SIL-20AC XR), degasser (DGU-20A3R), and column oven (CTO-20AC) were utilized for analysis. Chromatography separation was performed on an ACQUITY™ UPLC BEH C_18_ column (2.1 × 50 mm, 1.7 µm, Waters Corp., Milford, MA, USA).

### 4.3. UPLC–MS/MS Conditions

The mobile phase was composed of 0.1% formic acid and 5 mmol/L ammonium formate in deionized water (mobile phase A) and 0.1% formic acid in methanol: acetonitrile (20:80, *v/v*, mobile phase B). The gradient started with 25% B (4.0 min), then ran linearly to 60% B in 1 min and maintained for 0.5 min, next ramped to 80% B in 0.1 min and held for 0.4 min. Finally, the gradient returned to 25% B immediately and followed by a 1.0 min re-equilibration of the column. The flow rate was 0.4 mL/min, and the total run time was 7.0 min. The column temperature was set to 40 °C and the autosampler temperature was set to 10 °C, with an injection volume of 20 μL for plasma and 5 μL for urine. The positive electrospray ionization (ESI+) source and gas parameters were as follows: Temperature 550 °C; Ion Source Gas 1, 55; Ion Source Gas 2, 55; Curtain Gas, 35; Collision Gas, 7; and the Ionspray Voltage was 5500 V. Acquisition was performed in multiple reaction monitoring (MRM) mode. The ion transition and ionization conditions were optimized for maximum response. Data acquisition and analysis was operated on an Analyst software (version 1.7.1, AB SCIEX, Concord, Canada).

### 4.4. Stock Solutions, Calibration Standards and Quality Controls (QC)

The stock solutions (1 mg/mL) of apixaban, rivaroxaban, edoxaban, and their internal standards (IS) were prepared separately in acetonitrile and ultrapure water (50:50, *v/v*), while the stock solutions (1 mg/mL) of betrixaban and its internal standard were prepared individually in dimethyl sulfoxide (DMSO). The concentrations of standard calibration in human plasma (citric acid anticoagulation) samples were 1, 5, 10, 25, 50, 100, 250, and 500 ng/mL, while 10, 50, 100, 500, 1000, 2000, 5000, and 10,000 ng/mL in human urine for all analytes. The LLOQ, LQC, MQC, HQC, and DQC samples of all analytes were prepared at the concentrations of 1, 2, 40, 400, and 1250 ng/mL in the plasma samples, while 10, 30, 750, 7500, and 15,000 ng/mL in urine samples. The mixed internal standard solution was diluted to 200 ng/mL in acetonitrile. All stock solutions, calibration standards, IS, and QCs were stored at −80 °C until analysis.

### 4.5. Sample Preparation

Protein precipitation was employed to prepare the plasma. Briefly, an aliquot of 30 µL of mixed internal standard was added to a 1.5 mL centrifuge tube, then 30 µL of plasma and 60 µL acetonitrile were successively added to the centrifuge tube. After vortex mixing for 1.0 min, the tube was centrifuged at 17,000× *g* for 10 min; 40 µL of the supernatant was transferred to another centrifuge tube, then added 80 µL mobile phase A. Following this, the vortex mixed for 30 s and 20 μL of the mix solution was injected into the UPLC–MS/MS system for analysis.

For urine samples, 40 µL aliquot of mixed internal standard solution was added to 1.5 mL centrifuge tubes, followed by 20 µL of urine sample in a centrifuge tube, and then 940 µL of the acetonitrile-water (1:3, *v/v*). The centrifuge tube was vortex mixed for 30 s. Finally, a total of 5 µL of the supernatant was injected to the instrument system for analysis.

### 4.6. Method Validation

The method developed was validated according to the guidelines of the US Food and Drug Administration (FDA) (2018) [31], the European Medicines Agency (EMA) (2011) [32], the International Council for Harmonisation of Technical Requirements for Pharmaceuticals for Human Use (ICH M10) (2022) [33], and the Pharmacopoeia of the People’s Republic of China (2020) [34], including linearity, precision, accuracy, matrix effect, extraction recovery, stability, dilution integrity, and carryover.

#### 4.6.1. Linearity

For linearity, the peak area ratios of the analytes to their internal standards at eight levels were plotted against the nominal concentration (x) by least squares linear regression, with a weighting factor of 1/X^2^. The calibration curves with correlation coefficients R^2^ > 0.99 and less than ±15% deviation from the nominal value (while ±20% for LLOQ) were accepted.

#### 4.6.2. Accuracy and Precision

Six replicates of LLOQ, LQC, MQC, and HQC in plasma and urine were analyzed to assess the accuracy and precision of inter- and intra-day in three successive days. The precision was expressed as the relative standard deviation (RSD), while the accuracy was denoted as the relative error (RE). Both were required to be less than ±15% for LQC, MQC, and HQC, yet ±20% for LLOQ.

#### 4.6.3. Selectivity

Six independent blank plasma and urine samples were analyzed and compared with corresponding spiked LLOQ samples to evaluate the selectivity, and the peak areas of blank samples less than 20% of the LLOQ were acceptable.

#### 4.6.4. Matrix Effects and Recovery

The matrix effect was investigated through comparison of the peak area of the analytes in solutions to that of the analytes spiked at the concentrations of LQC, MQC, and HQC to six independent blank plasma extracts. The RSD of matrix effect at three concentrations was required to be less than 15%. The extraction recovery was obtained by contrasting the peak area from the regularly extracted three levels QC samples to that from the blank plasma spiked at the same concentrations after extraction, and the extraction recovery should be less than 115% for all levels QC with RSD less than 15%.

#### 4.6.5. Effect of Hemolysis and Hyperlipidemia on Plasma

The effect of hemolysis and hyperlipidemia on analytes quantification was investigated by preparing the corresponding LQC, MQC, and HQC samples using blank plasma separately containing 2% hemolysis and 300 mg/dl triglyceride; RE less than ±15% and RSD less than 15% were acceptable.

#### 4.6.6. Stability

For stability, six replicates of three levels QC (LQC, MQC, and HQC) were analyzed after the storage and preparation under certain conditions. The freeze-thaw stability was evaluated after three cycles of the freeze (−80 °C) and thaw (room temperature) before plasma and urine sample preparation. The short-term stability was assessed after the QC samples were kept at room temperature for 24 h in plasma and urine. The long-term stability (−80 °C for 112 days in plasma and 93 days in urine) was surveyed. The autosampler stability of the plasma and urine was tested after the processed QC samples were placed in an autosampler (10 °C) for 72 h. The reinjection stability of the plasma and urine was appraised by reinjection after storage at 10 °C for 69 h and 48 h, respectively. 

#### 4.6.7. Dilution Integrity

Six replicates of DQC (at 1250 ng/mL in plasma and 15,000 ng/mL in urine for all analytes) were 10-fold diluted with the pooled plasma or urine, then analyzed to assess the dilution integrity. The precision and accuracy should be less than 15%.

#### 4.6.8. Carryover

A blank sample after the highest calibration standard was analyzed to assess the method carryover. Furthermore, the carryover within 20% for all analytes and 5% for internal standard were acceptable.

### 4.7. Clinical Application

The study was conducted in accordance with the Declaration of Helsinki and approved by the Ethics Review Committee of Peking Union Medical College Hospital (the approval number, ZS-2796), and informed consent was obtained from all individuals included in this study.

Ultimately, a total of 18 plasma samples and 14 urine samples were collected from the patients treated regularly with rivaroxaban, apixaban, and edoxaban. For betrixaban, no plasma and urine sample from patients or healthy subjects was collected, since betrixaban was not on the market in China.

## 5. Conclusions

The generic UPLC–MS/MS methods for the simultaneous quantification of rivaroxaban, apixaban, betrixaban, and edoxaban in human plasma and urine were developed and validated to be accurate, reliable, precise, and sensitive. With the low labor-intensive sample preparation, high throughput, and rapid turnaround, the developed method was successfully applied to the patients and subjects for drug monitoring and assessment of anticoagulant activity.

## Figures and Tables

**Figure 1 molecules-28-02254-f001:**
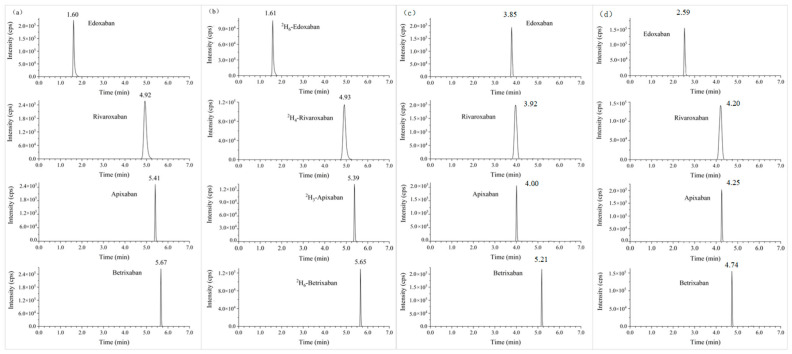
Typical extracted ion chromatograms of a processed blank plasma spiked with 50 ng/mL of selected DOACs (**a**) and their internal standards (**b**) on BEH C_18_, BEH Phenyl (**c**), and CSH C_18_ (**d**) columns under optimized conditions.

**Figure 2 molecules-28-02254-f002:**
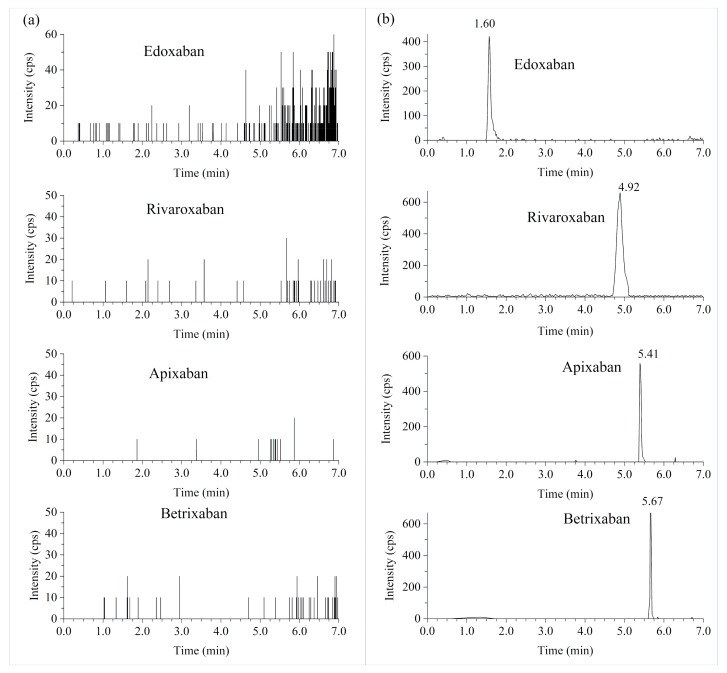
The UPLC/MS/MS chromatograms of extracted blank plasmas (**a**) and spiked with the lower limit of quantitation of 1 ng/mL in blank plasmas (**b**) of selected DOACs.

**Figure 3 molecules-28-02254-f003:**
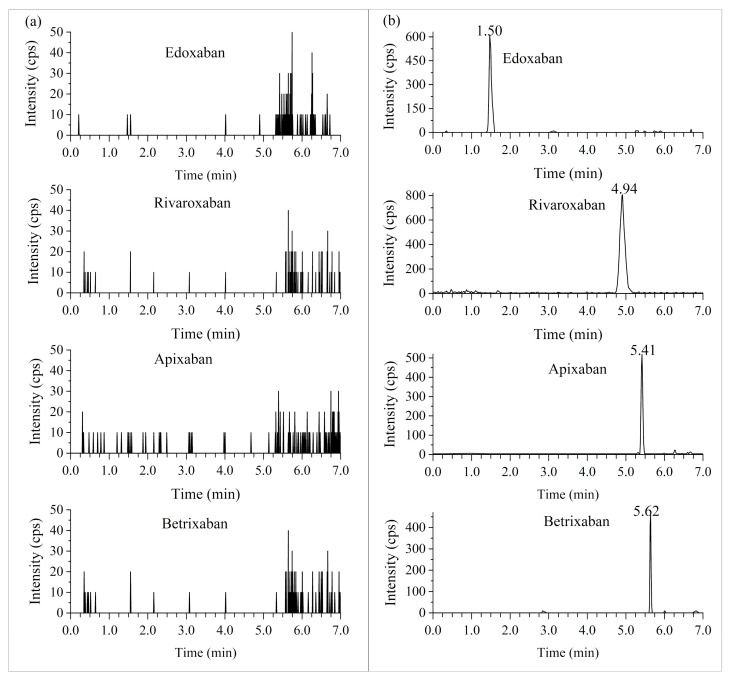
The UPLC/MS/MS chromatograms of processed blank urines (**a**) and spiked with the lower limit of quantitation of 10 ng/mL (**b**) in blank urines of selected DOACs.

**Figure 4 molecules-28-02254-f004:**
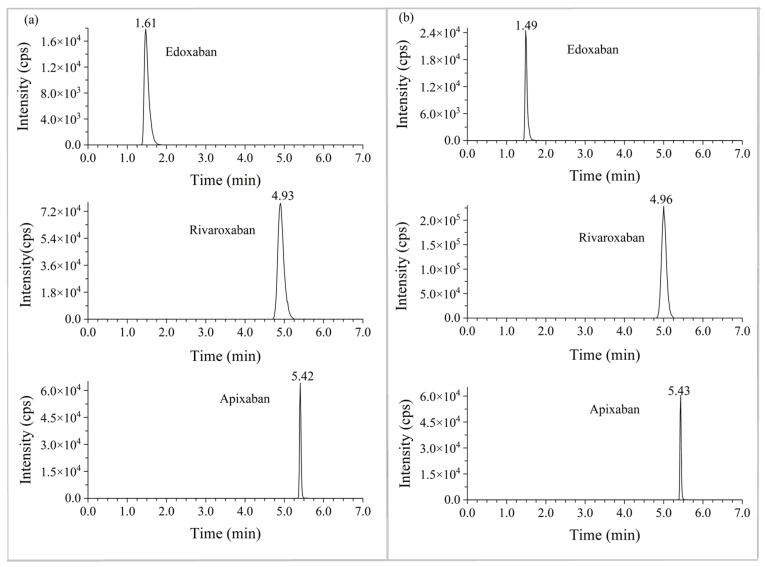
The UPLC/MS/MS chromatograms of the selected DOACs in the plasma (**a**) and urine (**b**) from clinical patients and subjects.

**Figure 5 molecules-28-02254-f005:**
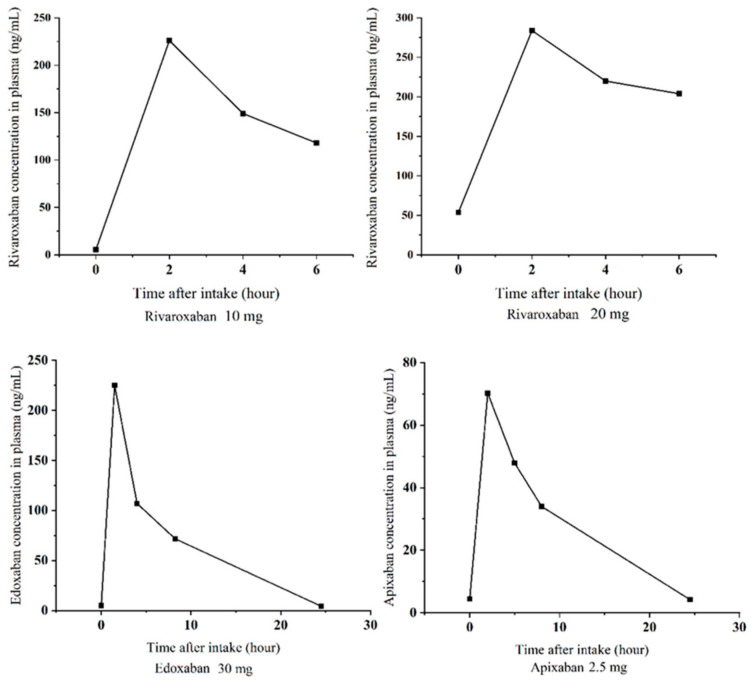
Brief concentrate-time curve of patients and subjects treated with rivaroxaban, edoxaban, and apixaban.

**Table 1 molecules-28-02254-t001:** Optimized transitions parameters for all analytes.

Compound	Retention Time (Minute)	Transitions (m/z)	DP (V)	EP (V)	CE (V)	CXP (V)
Rivaroxaban	4.92	436.0 > 145.0	168	7	40	15
^2^H_4_-Rivaroxaban	4.93	440.0 > 144.9	180	8	37	14
Betrixaban	5.67	452.1 > 324.2	90	8	55	18
^2^H_6_-Betrixaban	5.65	458.1 > 330.2	86	10	55	25
Apixaban	5.41	460.1 > 443.2	200	6	55	19
^2^H_3_-Apixaban	5.39	463.1 > 446.3	175	7	55	17
Edoxaban	1.60	548.1 > 366.3	57	6	26	25
^2^H_6_-Edoxaban	1.61	554.1 > 372.3	60	7	27	10

**Table 2 molecules-28-02254-t002:** Linearity for all analytes.

Matrix	Compound	Range	Calibration Curves	R^2^
Plasma	Rivaroxaban	1–500 ng/mL	Y = 0.00604X + 0.00198	0.9992
	Betrixaban	1–500 ng/mL	Y = 0.00529X + 0.00228	0.9990
	Apixaban	1–500 ng/mL	Y = 0.00403X + 0.00119	0.9986
	Edoxaban	1–500 ng/mL	Y = 0.0052X + 0.000502	0.9991
Urine	Rivaroxaban	10–10,000 ng/mL	Y = 0.00270X + 0.00440	0.9992
	Betrixaban	10–10,000 ng/mL	Y = 0.00208X + 0.00333	0.9956
	Apixaban	10–10,000 ng/mL	Y = 0.00313X + 0.00308	0.9993
	Edoxaban	10–10,000 ng/mL	Y = 0.00234X + 0.00648	0.9992

**Table 3 molecules-28-02254-t003:** Intra-day and inter-day accuracies and precisions of all analytes.

Matrix	Run Batch	Analyte	LLOQ		LQC		MQC		HQC	
			RSD (%)	RE (%)	RSD (%)	RE (%)	RSD (%)	RE (%)	RSD (%)	RE (%)
Plasma	Intra-day 1 (*n* = 6)	Rivaroxaban	5.0	−11.3	3.0	1.7	1.2	6.8	2.2	6.0
		Apixaban	11.2	−4.0	7.6	4.3	2.3	8.6	2.1	6.0
		Edoxaban	14.7	−6.6	8.1	−0.8	2.8	7.7	2.9	7.0
		Betrixaban	9.6	−4.4	5.0	2.4	2.5	8.3	1.6	4.0
	Intra-day 2 (*n* = 6)	Rivaroxaban	6.8	−3.9	3.4	2.6	2.2	3.8	1.2	−0.4
		Apixaban	12.8	6.1	6.8	9.3	2.1	8.8	1.2	0.5
		Edoxaban	10.6	7.5	5.7	4.3	1.4	5.7	2	2.8
		Betrixaban	7.9	−2.9	7.1	1.3	2.8	5.3	1.9	−2.1
	Intra-day 3 (*n* = 6)	Rivaroxaban	4.4	4.3	4.8	6.8	1.8	4.6	1	−2.4
		Apixaban	5.5	0.3	2.8	7.4	1.3	3.8	1.4	−5.3
		Edoxaban	10.6	−8.8	3.7	4.3	1	4.5	1.8	−0.9
		Betrixaban	6.3	−13.8	6	1.9	2.2	3.7	0.7	−3.2
	Inter-day (*n* = 18)	Rivaroxaban	8.5	−3.6	4.2	3.7	2.1	5.1	3.9	1.1
		Apixaban	11	0.8	6.0	7.0	3.0	7.1	5.0	0.4
		Edoxaban	14	−2.6	6.0	2.6	2.0	6.0	4.0	3.0
		Betrixaban	9	−7	6.0	1.9	3.0	5.8	4.0	−0.4
Urine	Intra-day 1 (*n* = 6)	Rivaroxaban	7.0	1.4	2.0	2.0	2.9	−0.3	2.3	−4.2
		Apixaban	11.1	2.7	3.6	8.3	1.2	3.7	1.5	−10.0
		Edoxaban	10.9	0.2	5.5	−5.9	2.0	−0.4	2.6	0.8
		Betrixaban	8.8	−9.8	6.1	5.6	3.7	−0.8	1.9	−2.0
	Intra-day 2 (*n* = 6)	Rivaroxaban	9.2	-8.0	5.7	0.8	1.8	1.8	1.2	−2.0
		Apixaban	9.2	2.1	4.4	5.3	3.9	3.8	1.2	−10.2
		Edoxaban	10.9	0.2	6.8	−0.7	2.0	−1.5	2.1	0.2
		Betrixaban	8.1	−10.0	4.8	5.4	3.6	4.8	3.4	0.0
	Intra-day 3 (*n* = 6)	Rivaroxaban	5.1	−4.9	2.8	3.1	1.9	3.6	1.3	−2.6
		Apixaban	6.9	0.1	5.1	7.3	2.8	10.4	5.1	−11.8
		Edoxaban	14.2	−5.1	4.6	5.3	3.9	5.0	4.1	1.4
		Betrixaban	10.3	−4.9	4.1	0.1	2.5	1.8	1.4	−1.0
	Inter-day (*n* = 18)	Rivaroxaban	8.0	−3.8	3.7	1.9	2.7	1.7	1.8	−2.9
		Apixaban	8.8	1.7	4.3	7.0	4.1	6.0	3.0	−10.7
		Edoxaban	11.6	−1.5	7.1	0.4	3.9	1.1	2.9	0.8
		Betrixaban	9.0	−8.1	5.4	3.7	3.9	1.9	2.0	−1.0

**Table 4 molecules-28-02254-t004:** The matrix effects and recovery of all analytes.

Matrix	Item	Analytes	LQC	MQC	HQC	RSD (%)
Plasma	Matrix effects (%)	Rivaroxaban	94.1	89.1	93.9	3.1
		Apixaban	87.5	90.9	95.6	4.4
		Edoxaban	86.5	92.5	96.6	5.5
		Betrixaban	97.5	92.2	96.8	3.0
	Recovery (%)	Rivaroxaban	101.3	99.1	97.5	1.9
		Apixaban	104.7	99.0	93.5	5.6
		Edoxaban	99.8	92.4	93.7	4.2
		Betrixaban	94.8	98.4	95.1	2.1
Urine	Matrix effects (%)	Rivaroxaban	101.9	101.1	101.1	0.4
		Apixaban	100.3	97.9	98.0	1.4
		Edoxaban	98.6	97.5	102.6	2.7
		Betrixaban	97.0	97.6	100.3	1.8
	Recovery (%)	Rivaroxaban	93.5	93.3	99.5	3.7
		Apixaban	92.5	97.2	98.6	3.3
		Edoxaban	94.4	95.0	97.8	1.9
		Betrixaban	85.1	87.2	95.9	6.4

**Table 5 molecules-28-02254-t005:** Effect of hemolysis and hyperlipidemia in plasma on analytes quantification.

Item	Analyte	LQC		MQC		HQC	
		RE (%)	RSD (%)	RE (%)	RSD (%)	RE (%)	RSD (%)
Hyperlipidemia stability (300 mg/dL)	Rivaroxaban	9.2	1.8	2.5	1.7	−1.9	1.2
	Apixaban	1.7	4.3	5.5	1.3	−2.1	2.4
	Edoxaban	9.8	4.7	8.7	1.8	4.2	1.6
	Betrixaban	−3.0	5.5	5.7	2.2	−1.4	1.8
Hemolysis stability (2%)	Rivaroxaban	2.1	2.2	−1.9	1.9	−3.8	1.3
	Apixaban	2.0	7.2	2.1	2.9	−3.3	1.1
	Edoxaban	8.9	5.4	2.4	2.5	2.7	1.2
	Betrixaban	0.1	8.5	−1.1	2.4	−3.0	2.0

**Table 6 molecules-28-02254-t006:** Stability of all analytes under different conditions in plasma.

Item	Analyte	LQC		MQC		HQC	
		RE (%)	RSD (%)	RE (%)	RSD (%)	RE (%)	RSD (%)
Reinject stability (10 °C for 69 h)	Rivaroxaban	5.8	3.9	7.8	2.8	7.3	2.2
	Apixaban	4.8	2.2	8.3	1.5	2.0	2.7
	Edoxaban	9.3	5.9	7.0	1.9	7.5	2.2
	Betrixaban	5.2	6.9	9.5	1.6	5.5	1.9
Autosampler stability (10 °C for 72 h)	Rivaroxaban	2.3	1.6	1.9	2.7	−2.6	1.5
	Apixaban	−3.3	3.9	3.7	1.9	−6.0	0.9
	Edoxaban	−0.3	4.5	1.3	1.8	−0.8	0.9
	Betrixaban	1.2	2.8	1.6	2.7	−4.5	1.4
Short time stability (room temperature for 24 h)	Rivaroxaban	−3.5	4.1	2.2	1.7	−2	1.3
	Apixaban	−3.1	6.3	3.7	1.8	−4.9	1.1
	Edoxaban	1.1	6.4	2.3	2.0	−1.6	2.0
	Betrixaban	0.1	3.1	1.4	2.3	−4.9	1.5
Freeze and thaw stability (−80 °C to room temperature, 3 times)	Rivaroxaban	−8.2	7.3	5.8	2.1	3.7	2.9
	Apixaban	0.8	5.0	5.2	2.6	−0.7	3.5
	Edoxaban	−5.2	4.2	1.2	1.8	−1.1	1.9
	Betrixaban	2.4	6.0	6.7	2.5	0.4	1.9
Long time stability (−80 °C for 112 days)	Rivaroxaban	10.8	2.5	7.0	2.6	0.8	0.7
	Apixaban	9.4	2.0	8.7	2.0	−0.8	0.6
	Edoxaban	8.2	4.3	7.3	1.9	1.7	1.0
	Betrixaban	4.0	5.7	7.3	2.6	−1.1	1.3

**Table 7 molecules-28-02254-t007:** Stability of all analytes under different conditions in urine.

Item	Analyte	LQC		MQC		HQC	
		RE (%)	RSD (%)	RE (%)	RSD (%)	RE (%)	RSD (%)
Reinject stability (10 °C 48 h)	Rivaroxaban	1.4	3.3	1.9	2.7	−1.9	1.9
	Apixaban	5.3	8.6	7.8	3.7	−5.5	2.0
	Edoxaban	−1.5	10.2	−1.5	2.7	−0.9	1.4
	Betrixaban	0.0	4.6	1.0	3.0	−2.1	3.2
Autosampler stability (10 °C for 72 h)	Rivaroxaban	6.2	3.2	3.7	3.0	−3.2	0.7
	Apixaban	3.0	8.9	7.5	2.7	−10.9	3.5
	Edoxaban	4.6	4.2	2.8	1.7	−0.5	2.2
	Betrixaban	−1.9	8.5	4.5	1.6	−1.1	2.1
Short time stability (room temperature for 24 h)	Rivaroxaban	1.0	3.5	0.8	2.0	−3.0	1.6
	Apixaban	10.0	5.7	1.5	3.0	−13.0	1.2
	Edoxaban	−1.7	4.8	−1.3	1.8	−0.8	1.5
	Betrixaban	2.7	12.3	1.9	2.9	−2.6	2.0
Freeze and thaw stability (−80 °C to room temperature, 3 cycles)	Rivaroxaban	−0.8	3.0	−3.1	2.2	−9.2	2.0
	Apixaban	5.6	6.6	5.0	3.6	−9.8	2.0
	Edoxaban	−4.6	3.6	−5.6	3.0	−6.8	2.2
	Betrixaban	−6.5	9.8	2.5	2.8	−0.9	3.9
Long time stability (−80 °C for 93 days)	Rivaroxaban	0.6	2.8	0.7	2.7	−7.7	3.7
	Apixaban	−2.5	5.8	−0.3	6.2	−8.2	6.3
	Edoxaban	−6.0	3.9	−0.5	5.2	−2.6	4.4
	Betrixaban	−1.4	9.6	−0.3	4.9	−2.1	4.7

## Data Availability

Data are available on request due to restrictions such as privacy or ethics.

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
