# Peer review of "Generic Methods for Simultaneous Analysis of Four Direct Oral Anticoagulants in Human Plasma and Urine by Ultra-High Performance Liquid Chromatography-Tandem Mass Spectrometry"

_molecules, 2023, doi:10.3390/molecules28052254_

Round 1
Reviewer 1 Report
The manuscript by Ren et al (Molecules) describes a rapid method for analysis of 4 oral coagulants in plasma and urine following conventional validation guidelines. Overall the bioanalytical approach utilized for method validation is standard and demonstrates its utility. However, several areas of the manuscript require grammatical improvement.
Minor Concerns:
1. The first sentence of the abstract does not make sense and needs to be rewritten.
2. The 2 sentences between line 52-55 need to be rewritten for clarity.
3. Line 61-62 – the authors state that BEH C18 columns performed the best, however, no data is presented to demonstrate that. The authors should consider showing the comparison for the different columns or methods evaluated.
4. Table 6 – (2nd row) – Do the authors mean ‘reinject stability 10 degrees C for 48 hours’ or is it 69 hours as written? That would make more sense in comparison to the other tables.
5. Line 172-177 needs restructuring for grammar
6. Line 192-197 needs grammatical restructuring
Author Response
Dear editors and reviewers,
Thank you very much for your suggestions. All your suggestions have very important guiding significance for our paper writing and scientific research work. We have marked all modifications with a green underline. The details are as follows:
- The first sentence of the abstract does not make sense and needs to be rewritten.
Response:
We have rewritten the first sentence of the abstract:
“The new direct oral anticoagulants (DOACs) are increasingly used to treat and prevent
thromboembolic disorders, monitoring concentrations may be valuable in some special scenarios to prevent clinical adverse events.”
- The 2 sentences between line 52-55 need to be rewritten for clarity.
Response:
Since line 52-55 are results and do not belong in the introduction., we chose to remove this part according to the suggestion of Reviewer 3.
- Line 61-62 - the authors state that BEH C18 columns performed the best, however, no data is presented to demonstrate that. The authors should consider showing the comparison for the different columns or methods evaluated.
Response:
We have added related content in Figure 1.
- Table 6 - (2nd row)- Do the authors mean 'reinject stability 10 degrees C for 48 hours' or is it 69 hours as written? That would make more sense in comparison to the other tables.
Response:
We have corrected this typo.
- Line 172-177 needs restructuring for grammar.
Response:
We have restructured line 172-177:
In 2012, the British Committee for Standards in Haematology (BCSH) published guidance for dabigatran and rivaroxaban therapy, patients taking rivaroxaban within 24 hours before surgery, patients with renal failure, patients with bleeding, patients with overdose, and patients with thrombosis were recommended to monitor drug concentrations during treatment.
- Line 192-197 needs grammatical restructuring.
Response:
We have restructured line 192-197:
In the multi-analyte approaches developed by Kathrin et al[6] and Vítězslav et al[25], the chromatographic peaks of apixaban, rivaroxaban and edoxaban were completely overlapping, which may be accounted for the unreliable quantification. However, in this study, baseline separation of all analytes was evidently observed in Figure 1.
Best Wishes
Sincerely,
Xiao-Hong Han, Xin Zheng and Jian-Wei Ren

Reviewer 2 Report
This is an interesting study aimed at developing multiple-analyte techniques of Direct oral anticoagulants for patients suffering from stroke or blood coagulation issues. This technique is innovative and monitoring the DOAC is very important. The authors in this study have come up with monitoring the effect of a group of anticoagulant drugs which seems promising. The authors have done decent background study as it looks well-referenced. Also the tables and figures are easy to follow. I would prefer little more explanation in the figure legends about the figure. The language also is flawless and does not need any fixing. Overall I am quite happy with the manuscript and would recommend acceptance post some minor corrections. I have attached my comments in the file.

Author Response
Dear editors and reviewers,
Thank you very much for your suggestions. All your suggestions have very important guiding significance for our paper writing and scientific research work. We have marked all modifications with a green underline. The details are as follows:
For reviewer 2
Comments and Suggestions for Authors:
This is an interesting study aimed at developing multiple-analyte techniques of Direct oral anticoagulants for patients suffering from stroke or blood coagulation issues. This technique is innovative and monitoring the DOAC is very important. The authors in this study have come up with monitoring the effect of a group of anticoagulant drugs which seems promising. The authors have done decent backaround study as it looks well-referenced. Also the tables and figures are easy to follow. I would prefer little more explanation in the figure legends about the figure. The language also is flawless and does not need any fixing. Overall I am quite happy with the manuscript and would recommend acceptance post some minor corrections. I have attached my comments in the file.
Response:
We have added some explanations in the legends of the figures, and made corrections based on the comments attached to the file.
Best Wishes
Sincerely,
Xiao-Hong Han, Xin Zheng and Jian-Wei Ren

Reviewer 3 Report
Dear Authors.
The proposed manuscript presents a very interesting work regarding the monitoring of four oral anticoagulants in urine and plasma using minimal sample preparation and ultra-high performance liquid chromatography-tandem mass spectrometry.
The work is scientifically sound and presents new data in this regard.
I believe that, after minor revision, it is suitable to be published in Molecules.
Specific remarks:
· The title is very extensive. Please shorten it.
· Lines 37-38: What special scenarios?
· Lines 52-57: These are results and do not belong in the introduction
· Correct every abbreviation to its full content the first time that they appear in the manuscript. E.g.: IS (line 67); LLOQ (line 91), RSD and RE (line 99), LQC, MQC and HQC (table 3)
Author Response
Dear editors and reviewers,
Thank you very much for your suggestions. All your suggestions have very important guiding significance for our paper writing and scientific research work. We have marked all modifications with a green underline. The details are as follows:
For reviewer 3
- The title is very extensive. Please shorten it.
Response:
We have shortened the title: Generic methods for simultaneous analysis of four direct oral anticoagulants in human plasma and urine by ultra-high performance liquid chromatography-tandem mass spectrometry.
- Lines 37-38: what special scenarios?
Response:
We have listed several special scenes in lines 35-36.
- Lines 52-57: These are results and do not belong in the introduction.
Response:
We have removed this section from the introduction.
- Correct every abbreviation to its full content the first time that they appear in the manuscript. E.g.: IS (line 67); LLOQ (line 91), RSD and RE (line 99), LQC, MQC and HQC (table 3)
Response:
We have checked and corrected the abbreviations in the full text.
Thanks again for your advice and taking the time to review our manuscript, we hope that the revised manuscript could meet your requirements
Best Wishes
Sincerely,
Xiao-Hong Han, Xin Zheng and Jian-Wei Ren
